# Polarization-Encoded Fully-Phase Encryption Using Transport-of-Intensity Equation

**Alok K. Gupta [1], Praveen Kumar [1], Naveen K. Nishchal [1,*] and Ayman Alfalou [2]**

1. Department of Physics, Indian Institute of Technology Patna, Patna 801106, India; alok.pph16@iitp.ac.in (A.K.G.); praveen.pph17@iitp.ac.in (P.K.)
2. LSL Teams, L@bISEN, Yncrea Ouest, 20 rue Cuirasse Bretagne, CEDEX 2, 29228 Brest, France; ayman.al-falou@isen-ouest.yncrea.fr
* Correspondence: nkn@iitp.ac.in

**Abstract:** In this study, we propose a novel method to encrypt fully-phase information combining the concepts of the transport of intensity equation and spatially variant polarization encoding. The transport of intensity equation is a non-iterative and non-interferometric phase-retrieval method which recovers the phase information from defocused intensities. Spatially variant polarization encoding employs defocused intensity measurements. The proposed cryptosystem uses a two-step optical experimentation process—primarily, a simple set-up for defocused intensities recording for phase retrieval and then a set-up for encoding. Strong security, convenient intensity-based measurements, and noise-free decryption are the main features of the proposed method. The simulation results have been presented in support of the proposed idea. However, the TIE section of the cryptosystem, as of now, has been experimentally demonstrated for micro-lens.

**Keywords:** optical encryption; transport of intensity equation; spatially variant polarized beam; phase-only information

## 1. Introduction

Information security has become one of the prime concerns in the current digital era, which has led to the development of many encryption–decryption techniques [1–5]. In optical cryptography, different features of light are utilized for the secure storage and transfer of information. Optical methods provide high-speed processing due to inherent parallelism and various degrees of freedom such as amplitude, phase, polarization, and orbital angular momentum, along with their spatial and temporal variations. For this reason, optical techniques of image encryption and decryption have found widespread interest among researchers [2,3]. Following this, several features of light have been explored to improve the performance and level of security [6–10]. The double-random phase-encoding (DRPE) scheme is one of the earliest reported encryption techniques, which serves as the main framework for optical cryptography. In DRPE and its variant techniques, phase encoding has been utilized where the information is encoded into the phase-only distribution of the optical field [11–14]. Other encryption methods based on various transforms such as the Fourier transform, Fresnel transform, fractional Fourier transform, gyrator transform, and wavelet transform have also been reported [15–20]. Earlier reported symmetric encryption schemes, including DRPE, were found to be vulnerable to specific attacks, which led to the development of asymmetric cryptosystems [21–23]. Asymmetric schemes based on phase truncation Fourier transform and the equal modulus decomposition techniques were also reported [3,24–26]. In these schemes, to achieve a higher level of security, decryption keys were generated during encryption and the use of random amplitude or phase masks as the encryption keys. In an encryption scheme, keys and encrypted information with high randomness are considered to be more secure. In cryptography, other techniques

such as image scrambling are also used to enhance randomness that ultimately improves security [27].

For convenience in optical implementation, encryption systems based on intensity-based measurements are desired since the recording and storage of intensity distribution is easier than phase [1]. Due to this reason, the polarization of light has been extensively explored in optical cryptography since the polarization distribution of the optical field can be characterized through intensity measurements using Stokes polarimetry. In recent years, optical fields with spatially variant polarization (SVP) have been utilized in cryptography to achieve higher encoding capacity. Such beams have SVP across the optical beam cross-section [28–31].

Along with intensity distribution, the phase information of any object or sample carries vital information. Therefore, attempts have been made to develop schemes for the encryption of phase information which have been referred to as fully-phase encryption [32–37]. Such encryption schemes have been reported as more tolerant to noise and offer a higher level of security [3]. Efficient phase retrieval techniques are crucial for such schemes, which have been widely investigated. The phase retrieval techniques can be categorized into interferometric and non-interferometric methods. Interferometric techniques suffer from problems such as vibration isolation, the requirement of coherent source, and phase unwrapping [3,34]. Non-interferometric methods can again be classified as iterative and non-iterative methods. Iterative methods have associated complexities, such as that they rely on a perfect coherent light source. Therefore, phase aberration and the coherent noise problem usually prevent accurate and high-quality phase retrieval [36,37]. The transport of intensity equation (TIE) is a non-iterative and non-interferometric phase-retrieval technique. It is a two-dimensional (2D) Poisson equation between intensity and phase [38,39]. As compared to the interferometric methods, TIE has the advantage of retrieving the phase information directly from intensity recordings which significantly simplifies the optical setup [40–42]. Recently, TIE-based image encryption schemes have been reported, but they used amplitude input data with a random amplitude mask in an asymmetric approach [43] and for authentication [44].

The research hypothesis of this study can be stated as follows. The manuscript demonstrates a method to encrypt fully-phase information using the transport of intensity equation and stokes polarimetry [34,35,45]. The polarization of light has been extensively explored in optical cryptography since the polarization distribution of the optical field can be characterized through intensity measurements using Stokes polarimetry [28–31]. The Arnold's cat map-based image scrambling algorithm has been used to randomize the information to enhance robustness [27]. As compared to the previous schemes based on TIE [43,44], the proposed technique has the advantage of fully-phase encryption with enhanced security [45]. Along with robustness in the proposed security system, the method has the benefit of intensity-based measurements for phase retrieval and ciphertext generation, unlike the existing DRPE. Our method has the advantage of using spatially variant polarized beam and stokes polarimetry in addition to TIE phase imaging, which adds polarization as an additional degree of freedom apart from the amplitude and phase. This is the major advantage of the current method over existing TIE-based cryptosystems [43,44].

The rest of the paper is organized as follows. First of all, related work has been briefly explained in Section 2. Section 3 discusses the principle of the proposed encryption scheme. The method to optically implement the proposed scheme is described in Section 4. Section 5 contains the simulation results and discussion. The performance and security analysis have been discussed in Section 6. Section 7 presents the conclusion.

## 2. Related Work

The technology for information security using digital methods is being enhanced by applying more powerful algorithms. Longer key lengths are chosen such that current computers using the best cipher-cracking algorithms would require an unreasonable amount of time to break the key. The mathematical theories play an important role in digital encryption. The advanced encryption standard (AES) is an encryption standard adopted

that is simpler and faster. It is symmetric-key cryptography, where two parties exchange the key in a secure way. In this case, the key is kept secure which causes great difficulty for the parties. The Rivest, Shamir, and Adleman (RSA) is another digital encryption scheme that uses public-key cryptography. It works on at least two different keys, i.e., public key and private key [1–3].

It is believed that optical encryption methods would provide a more robust environment that could be more resistant to attacks compared with digital electronic systems. Owing to the speed of light, optical technologies have been extensively studied for information security, encryption and authentication. The optical wavefront consists of many degrees of freedom such as amplitude, phase, polarization, orbital angular momentum of photons and different types of transformations and multiplexing techniques, enhancing the encryption to make it more difficult to attack [3]. Quantum cryptography is another technique for transmitting secure information where the data's security is guaranteed by quantum mechanics principles [4,5]. The role of imaging and sensing is growing in information systems, and so the security of such systems becomes crucial. Biometrics, surveillance, inspection, medical, and health monitoring are all fields where security is quite important. In order to protect such data from theft, falsification, and counterfeiting, the application of optical cryptographic technology is recommended [6–10].

The common characteristics of every optical cryptosystem are effectively encoding, decoding and transfer of information. Due to this, the intensity-based ciphertext and key generation is more suitable. In DRPE-based optical encryption, the input information is encrypted by the amplitude of light modulated by random phase and Fourier transform. There are some variations of DRPE using Fresnel and fractional Fourier transforms. The complex ciphertext was the main disadvantage with the DRPE systems along with the speckle noise. Digital holography (DH) is also a powerful optical method used for encryption; therefore, there have been significant progresses in DH-based encryption methods over the last two decades. QR codes have also appeared as a new tool for optical encryption. However, the limited sizes of QR code and the presence of diffraction noise are the issues. New data containers can be designed to overcome these practical issues. New areas in optics, such as terahertz imaging, optical vortices, etc., can also accelerate the study of optical encryption [1–3].

The TIE is a non-interferometric and non-iterative phase recovery technique that uses the defocused intensities only. Due to its computational efficiency and experimental simplicity, it has been extensively studied in various applications, including information security [36–41]. The vector beams-based optical cryptosystems have several advantages. The intensity-based recordings benefit the storage and transmission of the encoded information. Moreover, the spatial variation of polarization enhances the encoding capacity [42,43]. Some recent work from our group has utilized the advantages in different novel encryption methods [44,45]. We are combining the advantages of the TIE and vector beam to enhance the experimental simplicity and encoding capacity with this novel optical cryptosystem. TIE has the advantage of retrieving the phase information with defocus intensities only through a simple experimental setup. On the other hand, the spatially variant polarization-based vector beam and stokes polarimetry measurement increase the robustness of the security measures.

## 3. Theoretical Analysis

### 3.1. Transport of Intensity Equation

TIE is a well-known method to retrieve phase information from multiple defocused intensity distributions [34,35]. The TIE is given by

$$\nabla_\perp [I_z(x,y) \cdot \nabla_\perp \phi_z(x,y)] = -\frac{2\pi}{\lambda}\frac{\partial I}{\partial z} \tag{1}$$

where $\lambda$ is the wavelength of the light source, $I_z(x,y)$ and $\phi_z(x,y)$ denote the intensity and phase distributions at a particular plane on the optical axis $z$, respectively. $\nabla_\perp$ denotes

the gradient operator in the transverse plane. The phase can be determined from the differentiation of intensities at several planes in the near field region. For relating directly to phase distributions, Equation (1) can be simplified as [36,37],

$$\nabla^2_\perp \phi(x,y) = -\frac{2\pi}{\lambda I_0}\frac{\partial I(x,y)}{\partial z} \tag{2}$$

which is the 2D Poisson equation and can be easily solved by applying the 2D fast Fourier transform algorithm to recover the phase distribution $\phi(x,y)$ of the object. The intensity is assumed to be constant around the in-focus plane, consistent with a very weakly absorbing or non-absorbing thin sample.

### 3.2. Principle of Proposed Cryptosystem

In this section, we discuss the principle of the proposed scheme for securing the phase distribution $\phi(x,y)$ of the object, which is referred to as the plaintext. The flowchart of the proposed scheme has been shown in Figure 1. At first, two defocused intensity distributions $I_1(x,y)$ and $I_2(x,y)$ are captured using the TIE optical setup while taking the phase object as a sample. The image scrambling algorithm is then applied to the defocused intensity distributions $I_1(x,y)$ and $I_2(x,y)$ to randomize the original organization of pixels. In this study, the Arnold's cat map-based scrambling algorithm has been used, which is a 2D invertible chaotic map [3]. After applying the algorithm, the scrambled intensity images are obtained, denoted as $I_{1s}(x,y)$ and $I_{2s}(x,y)$.

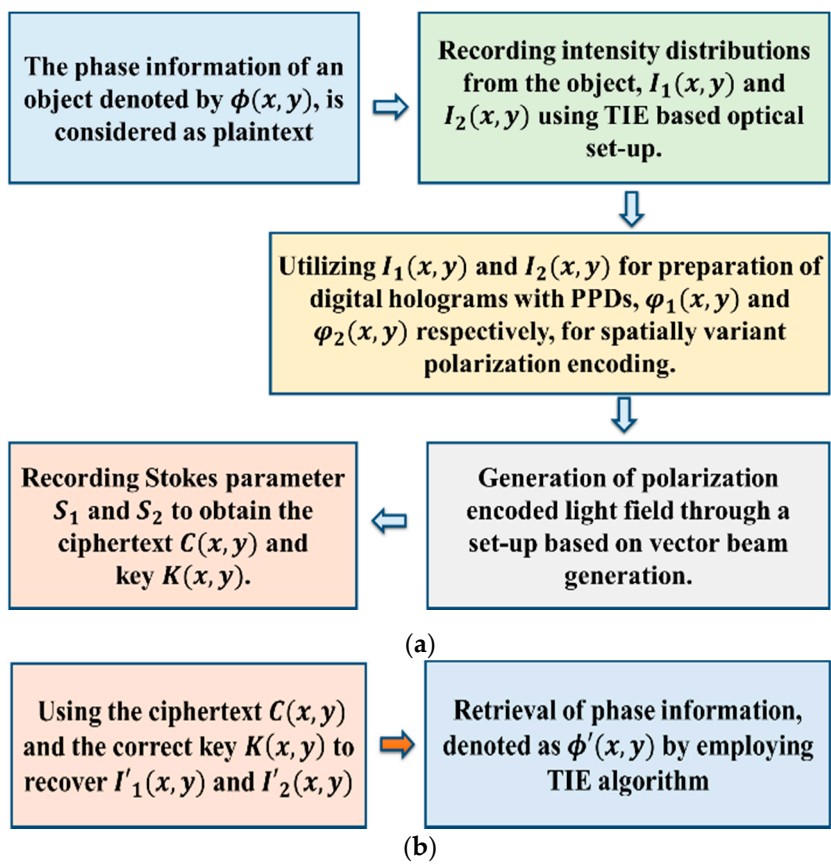

**Figure 1.** (**a**) Flowchart of proposed encryption and (**b**) decryption process.

The encryption is performed using a vector field generator setup. A vector field is an in-homogeneously polarized optical field where the polarization states are predominantly linear [28]. Vector fields can be obtained by the modulation of appropriate phase profile distributions (PPDs) into the orthogonal polarization components of an optical field. In

this case, PPDs $\varphi_1(x,y)$ and $\varphi_2(x,y)$ are evaluated from the scrambled intensities $I_{1s}(x,y)$ and $I_{2s}(x,y)$, respectively, by mapping their pixel values with the phase values within the range of 0 to $2\pi$. When these PPDs are modulated into orthogonal components of the light field in circular polarization basis, the resultant vector field can be represented in terms of the Jones matrix as [12],

$$E(u,v) = A\exp(i\varphi_1)\begin{pmatrix} 1 \\ -i \end{pmatrix} + A\exp(i\varphi_2)\begin{pmatrix} 1 \\ i \end{pmatrix} \tag{3}$$

where the $(1\ {-}i)$ and $(1\ i)$ represent the Jones vectors for left circularly polarized (LCP) and right circularly polarized (RCP) light wave, respectively. $A$ denotes the amplitude. The intensity distribution of the optical field represented by Equation (3) is recorded for the determination of Stokes parameters $S_1$ and $S_2$. These parameters of the above field are evaluated as [44,45],

$$\begin{aligned} S_1(x,y) &= 2A\mathrm{Im}(E_r E_l^*) = 2\sin(\varphi_2 - \varphi_1), \\ S_2(x,y) &= 2A\mathrm{Re}(E_r E_l^*) = 2\cos(\varphi_2 - \varphi_1). \end{aligned} \tag{4}$$

where $E_r$ and $E_l$ denote the RCP and LCP components, respectively. The ciphertext is evaluated using the Stokes parameters $S_1$ and $S_2$ using the following relation,

$$C(x,y) = \tan^{-1}(S_1/S_2) \tag{5}$$

whereas the key is generated numerically using one of the PPDs through the following relation,

$$K(x,y) = \tan^{-1}(S_1/S_2) - \varphi_1(x,y) \tag{6}$$

The decryption process consists of two steps, as illustrated in Figure 1b. For decryption, the authorized receiver must have the proper knowledge of the ciphertext $C(x,y)$ and the key $K$. In the first step, both PPDs are determined using the following relation:

$$\begin{aligned} \varphi_1{}'(x,y) &= C(x,y) - K(x,y) \\ \varphi_2{}'(x,y) &= 2C(x,y) - K(x,y) \end{aligned} \tag{7}$$

where $\varphi'_1(x,y)$ and $\varphi'_2(x,y)$ are the decrypted PPDs from which defocused intensity distributions $I_1(x,y)$ and $I_2(x,y)$ are recovered by applying the reverse of scrambling algorithm. In the final step, the decrypted phase distribution of object $\phi'(x,y)$ is evaluated using the TIE-based phase retrieval algorithm, as discussed in Section 2. The mathematical expression for phase recovery using the TIE method can be shown as follows:

$$\frac{\partial I(x,y)}{\partial z} = \frac{I_1(x,y) - I_2(x,y)}{2\Delta z} \tag{8}$$

$$\phi(x,y;z) = \Im^{-1}\left[ -k0\frac{\Im_{x,y}\left[ \frac{1}{I(x,y)}\frac{\partial I(x,y)}{\partial z} \right]}{k_x{}^2 + k_y{}^2} \right] \tag{9}$$

## 4. Methods

The implementation of the proposed scheme requires two optical setups, one for recording the defocused intensities of the phase object for TIE-based phase retrieval, while another setup is required for encrypting the information using polarization encoding. These optical setups have been discussed in the following sub-sections.

### 4.1. Setup for Intensity Recording for TIE-Based Phase Retrieval

The defocused intensity distributions for the TIE method can be easily recorded by a *4f* imaging setup, as shown in Figure 2. When the light beam illuminates onto a nearly transparent phase object and the intensity distribution of transmitted light can be recorded

at two slightly different positions which have been denoted as $I_1(x,y)$ and $I_2(x,y)$. In this study, we have used the micro-lens as a pure phase sample. A low-cost light-emitting diode (LED) (mean wavelength ~629 nm, FWHM ~ 13 nm) has been used as an illumination source. A biconvex lens is used after LED as a condenser lens. The sample has been placed on a mechanical stage. Then, a microscopic objective and a collimating lens make a magnified image on the image plane. The magnified image is relayed to the CCD camera (Imaging Source, number of pixels: 2592 × 1944, pixel size: 2.2 μm × 2.2 μm) through the *4f* imaging processor. The camera has been placed on a motorized translation stage (make: Thorlabs, Newton, NJ, USA) with a minimum achievable shift of 0.05 μm.

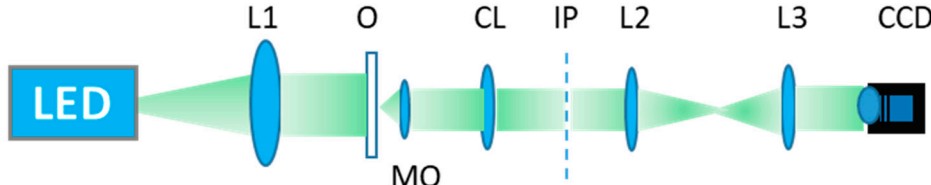

**Figure 2.** Schematic of the experimental setup for TIE; L1: Condenser lens, O: Object, MO: Microscopic objective lens, CL: Collimating lens, IP: Image plane, L2 and L3: *4f* imaging relay system, CCD: Charge-coupled device.

### 4.2. Setup for Polarization Encoding by Vector Field Generation

Different approaches and optical setups have been reported, which can be employed to generate a vector field, as described in Equation (3) [46,47]. The methods use spatial light modulators (SLMs) for encoding the PPDs into the orthogonal components of light through computer-generated holograms (CGHs). A single SLM-based optical setup can be utilized for implementing the proposed scheme. The schematic diagram of the setup is shown in Figure 3. In this case, the CGHs encoded onto the SLM consist of 2D holographic grating, a combination of two one-dimensional gratings aligned along *x*- and *y*-directions and each with their own phase distribution, namely $\varphi_1(x,y)$ and $\varphi_2(x,y)$, respectively. When linearly polarized light passes through the SLM and proceeds through a converging lens, diffraction orders are obtained. The first diffracted order (+1) generated along the *x*- and *y*-directions is isolated using the spatial filter and then passed through a quarter-wave plate to convert these two diffraction beams into right and left-hand circular polarization. Another grating can be used such that both these diffraction orders overlap and stay overlapped beyond that. The resultant is the vector field as illustrated in Equation (3). Stokes parameters can be evaluated by recording the intensity distributions of the resulting field to obtain the ciphertext.

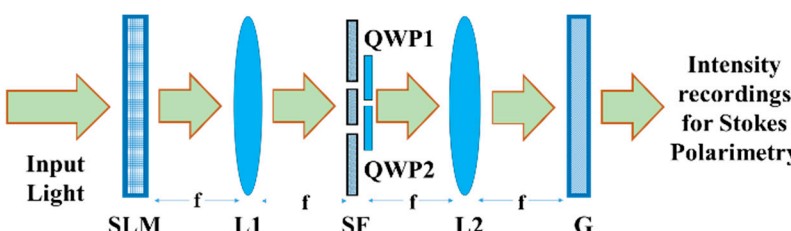

**Figure 3.** Schematic diagram for information encoding into SVP beam L: lens, f: focal length, G: grating.

The optical implementation seems to be a two-step cryptosystem at this time. There is good scope to improve the experimental setup. It can be made simpler once we do the experiments. In the TIE setup, the number of used lens can be reduced as per the requirement. Moreover, the vector beam generation can also be simpler with fewer optical components that do not use the grating [47].

## 5. Results and Discussion

The numerical simulation has been performed on MATLAB 2017b platform to verify the proposed encryption scheme while considering two different cases, which are discussed in subsequent sub-sections. In the first case, the phase distribution to be encrypted is in the form of a 2D image, while in the second case, plaintext is the information of phase delay introduced by an actual micro-lens. At this time, the TIE experimentation part has been done for the micro-lens phase sample. We are hopeful of doing the rest of the experiments in our future study.

### 5.1. 2D Phase Information

In this case, the proposed method has been verified by taking a phase-only function as the plaintext with unit amplitude. A real color picture taken in the laboratory has been used as an input image, as shown in Figure 4a. Its phase distribution $\phi(x,y)$ is in the form of a grayscale image with $256 \times 256$ pixels which has been used for further simulation. Fresnel-diffracted intensity distributions $I_1(x,y)$ and $I_2(x,y)$ are then obtained from the input phase-only function [30]. One of them is shown in Figure 4b. The scrambling algorithm is applied to the resultant intensity distributions and then converted into PPDs $\varphi_1(x,y)$ and $\varphi_2(x,y)$. One of the PPDs is shown in Figure 4c. One of the two diffracted images and their scrambled images are shown for the sake of brevity.

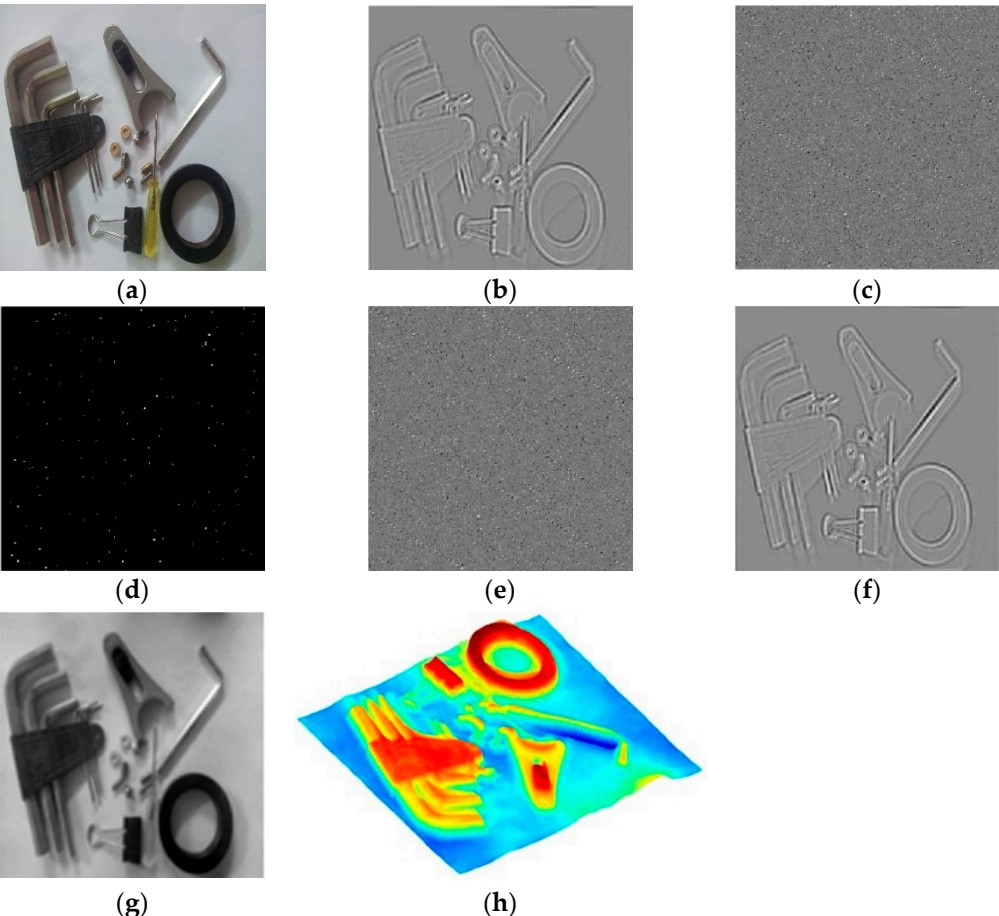

(a)  (b)  (c)

(d)  (e)  (f)

(g)  (h)

**Figure 4.** Results for a 2D image. (**a**) A picture of some lab components, (**b**) one of the two digitally propagated defocused intensities, (**c**) one of the scrambled images of the defocus intensities, (**d**) ciphertext, (**e**) key, (**f**) decrypted defocused intensity, (**g**) retrieved phase map, and (**h**) surface plot of the phase map.

The optical field is encoded with these PPDs to perform encryption, as discussed in the previous section. The vector field is evaluated through the process as illustrated in Equation (3). For these optical fields, the Stokes parameters $S_1$ and $S_2$ are evaluated using Equation (4). Ciphertext $C(x,y)$ is obtained using these parameters, as illustrated in Equation (5) and shown in Figure 4d. Similarly, using Equation (6), the generated key $K(x,y)$ is shown in Figure 4e. The decryption has been carried out with the correct keys as discussed. The PPDs are first obtained as illustrated in Equation (7), and the phase information $\phi'(x,y)$ is recovered using the TIE-based phase retrieval algorithm, as described in Section 2. Figure 4g shows the recovered phase information, and its three-dimensional (3D) plot is presented in Figure 4h.

### 5.2. 3D Phase Information

In the second case, the phase distribution of the micro-lens has been used as plaintext. The intensity distribution of the micro-lens has been captured through a TIE-based setup, as discussed in Section 4.1. The captured intensity distributions $I_1(x,y)$ and $I_2(x,y)$ are shown in Figure 5a,b, respectively. Afterwards, numerical simulation has been performed similarly to the previous case. The encryption has been performed as discussed through SVP encoding. The corresponding vector field is obtained using Equation (3), and the ciphertext $C(x,y)$ and key $K(x,y)$ are obtained using Stokes polarimetry, as illustrated in Equations (4) and (5). The ciphertext and key, in this case, have been shown in Figure 5d,e, respectively. The decryption is performed as described by using the correct key and the ciphertext. The decrypted phase distribution is shown in Figure 5g, and the corresponding 3D plot is shown in Figure 5h.

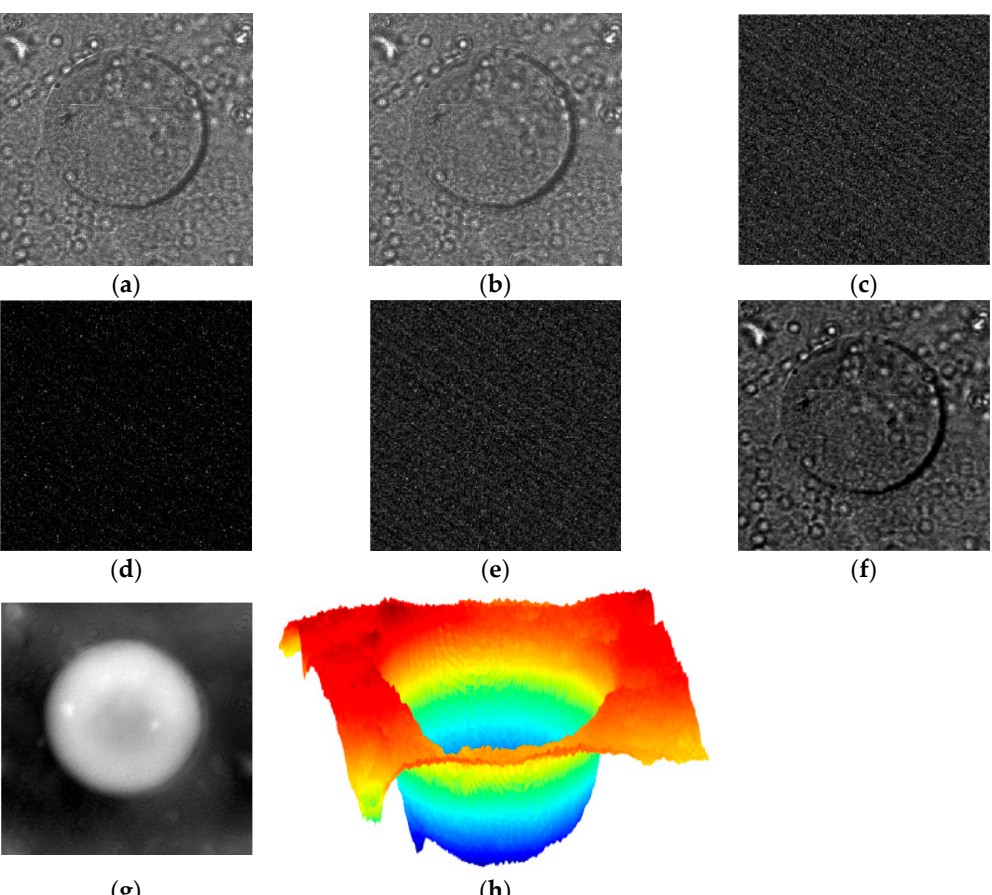

(a)          (b)          (c)

(d)          (e)          (f)

(g)          (h)

**Figure 5.** Results with a micro-lens used as sample. (**a**) and (**b**) Defocused intensities recorded on CCD camera, (**c**) one of the scrambled images of the defocus intensities, (**d**) ciphertext, (**e**) key, (**f**) decrypted defocused intensity, (**g**) retrieved phase map, and (**h**) surface plot of the phase map.

*5.3. Discussion*

When the fully phase image has been encrypted implemented optically, the system becomes difficult to replicate. Moreover, the phase image encryption techniques are also resistant to additive or multiplicative noise. In the literature, it was shown through DRPE-based approaches that the decrypted images from fully-phase encryption are more robust to noise than amplitude-based encryption [30,31]. Due to the importance of phase information encryption, we have shown the fully-phase encryption using the TIE. There are some other cryptosystems using the TIE principle in recent times [39,40]. Zhang et al. encrypted a phase-encoded plaintext using TIE and phase-truncated Fresnel diffraction. An amplitude-phase retrieval method has been used to decrypt the information in addition to the TIE [39]. In another work, Sui et al. demonstrated the multiple-image authentication using TIE. Similarly, the phase-encoded plaintext has been used with significant blocks chosen from the multiple plain images. The information has been encrypted by a random amplitude mask and Fresnel diffraction [40]. We have shown the results with the real pictures for plaintext (an image of lab components and a micro-lens image recorded using CMOS camera) as 2D and 3D information, unlike the previous TIE-based encryption studies. It is also important to mention that our study utilizes the phase (the phase information as a plaintext), amplitude (as ciphertext) and polarization (for information encoding) in a single cryptosystem, which makes it a complete optical cryptosystem. In the present scheme, the key size is proportional to the size of input information (plaintext). For example, in the results shown in Figure 4 of the manuscript, the size of both the key as well as plaintext remains $256 \times 256$ pixels. This is similar to most of the existing image encryption schemes. For instance, in the case of DRPE and Exclusive OR-encryption, random phase masks are used as keys that have the same size as plaintexts [1–3].

In the proposed method, the decryption key is a function of the input image since it has been obtained from the Stokes parameters S1 and S2, as stated in Equation (4). These parameters measure an important property of light, which has an infinite degree of freedom. A complex nonlinear relationship between the input image and the key therefore exists because of the introduction of real and imaginary terms, which greatly increase the key strength. Secondly, the key size is directly dependent on the size of the input image. The key size dynamically increases with the increase in size of the input image. Hence, a variable key with variable length would offer greater complexity as compared to methods which employ a fixed key with a fixed length.

## 6. Security Analysis

The security analysis of any proposed encryption scheme is very important, which has been discussed in this section. The immunity of a cryptosystem against attacks reflects the strength of the system. The brute force attack is one such widely used attack that is based on a random key search. If $N \times N$ is the number of pixels of a key and $M$ is the possible value of each pixel, the number of trials to retrieve the key will be $M^{N \times N}$ [3]. The number of trials further increases with the size of the used key. To substantiate this, we assume that the attacker has ciphertext as well as a priori information about the encryption algorithm. The attacker assumes random distributions as the key and then tries to decrypt the ciphertext with the assumed key and the known algorithm. For quantifying the decryption quality, the correlation coefficient (CC) of the decrypted image (using brute force) with the plaintext for each trial is evaluated. The decryption has been carried out using randomly generated keys. The plot of CC versus the number of trials for micro-lens is shown in Figure 6a. It can be seen that decryption remains unsuccessful even after 1000 decryption trials, and for all trials, the CC value remains below 0.02. Hence, it can be concluded that the attacker cannot retrieve the plaintext even after a large number of trials. It is also important to note that a known-plaintext attack does not apply to such a cryptosystem [12].

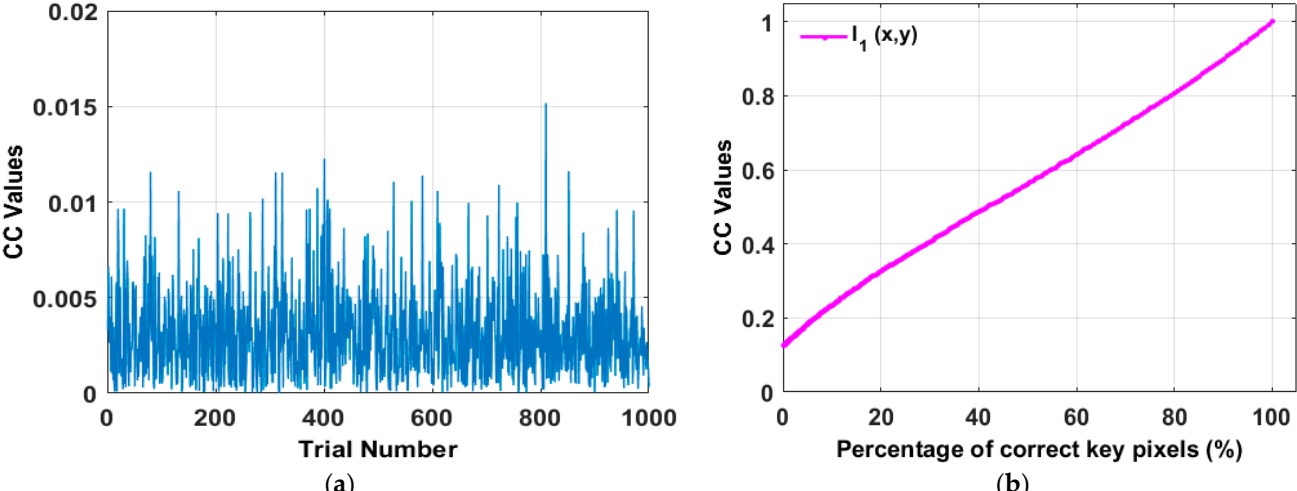

**Figure 6.** (**a**) The plot of CC versus the number of trials for micro-lens decryption trials using random keys, (**b**) the plot of CC values versus the percentage of correct key pixels.

The effect of occlusion has also been tested for the proposed scheme in the revised manuscript. In this case, the decryption is carried out by using partially correct keys. To test the originality of decrypted information, it is then compared with the input plaintext to obtain the CC values. The plot of obtained CC values at different percentages of correct key pixels is shown in Figure 6b. It shows the occlusion resistance of the proposed encryption scheme. For a secure encryption scheme, decryption keys should have sufficient randomness and large space. It makes it difficult to predict the geometry of the encryption resources. In the proposed scheme, the key-space is proportional to the size and pixel range of input information. The decryption keys are plaintext-dependent which automatically provide a one-time pad encryption mechanism, as in the case of asymmetric encryption schemes. Encryption schemes that use this protocol have the advantage of being resistant to known-plaintext attack, in which the attacker tries to find the keys with an already known ciphertext–plaintext pair [3]. Since the encryption key changes with the plaintext, the key retrieved by an attacker for a given plaintext–ciphertext pair would be of no use for other plaintexts.

In the present case, the encryption key can be retrieved from the known pair of plaintext-iphertext, only if the complete geometry of the encryption system is also known to the attacker. However, in this analysis, it is important to note that the encryption key is uniquely generated for each plaintext. Since the encryption key changes with the plaintext, the key retrieved by an attacker for a given plaintext–ciphertext pair would be of no use in decrypting another ciphertext. Our method of optical encryption is similar to asymmetric cryptography based on phase-truncated Fourier transforms which works with a one-time pad manner. This means that each encryption process will generate a new set of keys used for decryption [24]. Sufficient randomness and large space in the key and ciphertext are desirable to make it difficult for the attacker to predict the geometry of the encryption resources. This is why the Arnold's cat map-based randomization has been used in the proposed scheme.

In this case, the output (ciphertext) is affected by any small change in the input (plaintext), as the present method is based on the approach of pixel-by-pixel encryption. This feature is similar to existing cryptosystems based on spatially varying polarization [30,31]. In the present case, the scrambling algorithm has been applied, which makes the prediction of encryption components more difficult. Differential attack involves the study of differentiability between the two ciphertexts which are obtained by simply changing one of the pixel values of one of the plaintexts. Let us denote one of the plaintext as I1 and the image obtained on changing one of the pixel values as I2. The corresponding ciphertexts are denoted by C1 and C2. For a robust cryptosystem, there should be a substantial difference between C1 and C2, such

that distinguishability can be established between the original plaintext I1 and changed plaintext I2. Since the present method uses pixel-by-pixel encryption, each plaintext produces unique ciphertext.

The same input sub-image in different locations would produce a unique ciphertext and a key. In this case, the overall image would be encrypted, without any compromise with the security. The input image with patterns would produce similar random distributions. This is because the ciphertext is obtained from the randomized distribution of Stokes parameters, which have a well-defined range.

## 7. Conclusions

We demonstrate a novel optical asymmetric encryption method to secure the phase information of an object which uses a TIE-based phase retrieval technique with SVP beam encoding and stokes polarimetry. SVP encoding has been used to achieve strong security and high encoding capacity. The implementation of the proposed encryption system is convenient because both phase retrieval and ciphertext generation are based on intensity measurements. The simulation results verify the feasibility of the proposed encryption scheme. In this study, we have experimentally demonstrated the TIE part of the cryptosystem for the micro-lens. However, we hope for a complete experimental demonstration of the proposed cryptosystem in our future study. It is important to note that different features of the vector field used in this scheme can introduce new aspects to the cryptosystems.

**Author Contributions:** Conceptualization, A.K.G. and P.K.; Methodology, A.K.G.; Software, A.K.G. and P.K.; Validation, A.K.G., P.K., N.K.N. and A.A.; Formal Analysis, P.K.; Investigation, A.K.G.; Resources, N.K.N.; Data Curation, A.K.G.; Writing—Original Draft Preparation, A.K.G.; Writing—Review & Editing, A.K.G., P.K., N.K.N. and A.A.; Visualization, A.K.G.; Supervision, N.K.N. and A.A.; Project Administration, N.K.N.; Funding Acquisition, N.K.N. and A.A. All authors have read and agreed to the published version of the manuscript.

**Funding:** This research received no external funding.

**Acknowledgments:** The authors thank Avishek Kumar for some fruitful discussions.

**Conflicts of Interest:** The authors declare that there are no conflict of interest related to this article.

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
