# Peer review of "Polarization-Encoded Fully-Phase Encryption Using Transport-of-Intensity Equation"

_electronics, doi:10.3390/electronics10080969_

Round 1

Reviewer 1 Report

This paper proposes an encryption method by combining the concepts of transport of intensity equation and spatially-variant polarization encoding as an optical cryptography.

Since optical cryptography is not familiar to readers and is different from the classical cryptography such as AES, RSA, and so on., I recommend the authors to describe the reason of using optical cryptography, where they are used, and the difference between optical cryptography and the classical modern cryptography in the related work section.

The security analysis of the proposed encryption method is not enough. The number of trial is too low and the criteria for successful attack is not defined. I also strongly recommend the authors to deal with the performance comparison to previously proposed other methods. Especially, the proposed method needs to be compared with the existing literature with respect to executing time, cost, security level, and so on.

I feel that the current form of this paper is not satisfactory to be published in this journal.

Author Response

Authors' response to reviewer's comments are attached.

Reviewer 2 Report

In the introduction the authors presented an introduction to information security. The positioning of the research topic in theory is correct. The purpose of the study was not correctly defined. The authors write: "we propose a new method ..." Can the proposal of a new method be the aim of research? Alternatively, the goal may be to compare the new method with the old method or the standard method. I would even suggest a research hypothesis.
The structure of the work is not entirely correct. I would suggest separating the Methods and data section. In this section, the methods must be presented and described. The stages of the research can also be presented in sequence. Results and discussion can be presented later. Subsections can be created. I understand a discussion as referring to other studies after presenting own's research results. This was not in the reviewed article. The authors referred to other studies in the introduction. The conclusions are very general. The literature review is quite poor. In total, only 29 literature items were used. This is a bit too little for a high IF journal. The scientific discussion was therefore very limited.

Author Response

Authors' response to reviewers' comments are attached.

Round 2

Reviewer 2 Report

I still feel unsatisfied with the hypothesis. I understand the added description, but a hypothesis is a argument, eg Method A is better in terms of accuracy of the results obtained than Method B (standard methods). It is enough to write just such a hypothesis at the end of the added text.
The additions added are fine and basically refer to my comments in the previous review.

Author Response

Referee # 2

I still feel unsatisfied with the hypothesis. I understand the added description, but a hypothesis is a argument, eg Method A is better in terms of accuracy of the results obtained than Method B (standard methods). It is enough to write just such a hypothesis at the end of the added text.

The additions added are fine and basically refer to my comments in the previous review.

We thank the reviewer for providing the relevant suggestion. We have added the following texts in the hypothesis part of the introduction section.

“The proposed scheme uses spatially-variant polarization states of light for encoding and encryption which enables the extraction of ciphertext through Stokes polarimetry, thus simplifying the optical implementation of the scheme. The use of polarization for securing the TIE-enabled phase information adds more degree of freedom to the encryption system which enhances its overall security. These are the major advantages of the proposed method over existing TIE-based cryptosystems [43,44].”
